# Antimicrobial Therapy Duration for Bloodstream Infections Caused by *Pseudomonas aeruginosa* or *Acinetobacter baumannii-calcoaceticus complex*: A Retrospective Cohort Study

**DOI:** 10.3390/antibiotics12030538

**Published:** 2023-03-08

**Authors:** Rodrigo Douglas Rodrigues, Rebeca Carvalho Lacerda Garcia, Gabriel Almeida Bittencourt, Vicente Bouchet Waichel, Ester Carvalho Lacerda Garcia, Maria Helena Rigatto

**Affiliations:** 1Medical Sciences Post Graduation Program, Universidade Federal do Rio Grande do Sul, Porto Alegre 90035-903, Brazil; 2Medical School, Pontifícia Universidade Católica do Rio Grande do Sul, Porto Alegre 90619-900, Brazil; 3Medical School, Pontifícia Universidade Católica do Paraná, Curitiba 80215-901, Brazil; 4Medical School, Universidade Federal do Rio Grande do Sul, Porto Alegre 90035-903, Brazil; 5Infectious Disease Service, Hospital de Clínicas de Porto Alegre, Porto Alegre 90035-903, Brazil

**Keywords:** bloodstream infections, *Acinetobacter baumannii-calcoaceticus complex*, *Pseudomonas aeruginosa*, treatment duration

## Abstract

Background: Ideal therapy duration for *Pseudomonas aeruginosa* or *Acinetobacter baumannii-calcoaceticus complex* (ABC) bloodstream infections (BSI) is not defined, especially in the context of carbapenem resistance. In this study, we compared short- (≤7 days) and long-term (>7 days) antimicrobial therapy duration for these infections. Methods: We performed a retrospective cohort study in two tertiary-care hospitals in Porto Alegre, Brazil, from 2013 to 2019. Eligible patients aged ≥18 years were included and excluded for the following criteria: polymicrobial infections, treatment with non-susceptible antibiotics, complicated infections, or early mortality (<8 days of active antimicrobial therapy). The 30-day mortality risk was evaluated using a Cox regression model. Results: We included 237 BSI episodes, 51.5% caused by *ABC* and 48.5% by *Pseudomonas aeruginosa*. Short-term therapy was not associated with 30-day mortality, adjusted hazard ratio 1.01, 95% confidence interval 0.47–2.20, *p* = 0.98, when adjusted for Pitt score (*p* = 0.02), Charlson Comorbidity Index score (*p* < 0.01), and carbapenem resistance (*p* < 0.01). Among patients who survived, short-term therapy was associated with shorter hospital stay (*p* < 0.01). Results were maintained in the subgroups of BSI caused by carbapenem-resistant bacteria (*p* = 0.76), *ABC* (*p* = 0.61), and *Pseudomonas aeruginosa* (*p* = 0.39). Conclusions: Long-term therapies for non-complicated *Pseudomonas aeruginosa* and ABC BSI were not superior to short-term therapy for 30-day mortality.

## 1. Introduction

Bloodstream infections (BSIs) are associated with high morbidity and mortality [1,2]. Gram-negative bacteria are etiological agents in 50.2% of BSI cases in Europe, 53.7% in Latin America, and 36.3% in North America, with a significant increase in all regions in the last decade [3]. In Brazil, a recent multicentre study showed that BSI represents 27.5% of the healthcare-associated infections, being 80.3% acquired in the intensive care unit (ICU) setting [4]. *Enterobacterales* species are the predominant etiological agents of Gram-negative BSI, followed by *Acinetobacter baumannii-calcoaceticus complex* (ABC) and *Pseudomonas aeruginosa* [3,4] The ideal duration of treatment for Gram-negative bacteria remains undefined, especially in infections caused by multidrug resistant (MDR) pathogens. Guidelines recommend from 7 to 14 days of treatment for Gram-negative BSI, leaving shorter- or longer-term therapies to clinical discretion [5,6,7]. Recently, a consensus guidance recommended 7 days of therapy for uncomplicated Gram-negative BSI [8].

Therapy duration becomes more controversial in infections caused by non-fermenting Gram-negative bacteria, such as *Pseudomonas aeruginosa* and ABC, considering some specificities, such as resistance to multiple drugs, virulence and pathogenicity, scarce therapeutic options, and few and conflicting studies on optimal duration of therapy [9,10,11,12]. Most studies that evaluated duration of treatment in Gram-negative BSI included predominantly community-acquired infections, with a low frequency of non-fermenters and multi-resistant bacteria. In these studies, results showed no difference between short-term and long-term therapy [13,14]. Recently, three retrospective cohort studies found that short-term therapy was non-inferior to long-term therapy for *Pseudomonas aeruginosa* BSI [9,10,11]. On the contrary, a study evaluating carbapenem-resistant *Acinetobacter baumannii* BSI in cancer patients observed higher mortality in patients treated for less than 14 days [12].

Our study aimed to evaluate the impact of short-term antimicrobial therapies (≤7 days) compared to long-term therapies (>7 days) on 30-day and in-hospital mortality of patients treated for *Pseudomonas aeruginosa* or *ABC* BSI.

## 2. Results

We evaluated 721 BSI episodes caused by *Acinetobacter* spp. or *Pseudomonas* spp., from which we included 237 in the final analysis (Figure 1). Patients had a mean age of 61.6 ± 16.0 years and 122 (51.5%) of the 237 patients were male. We isolated ABC in 122 (51.5%) and *Pseudomonas aeruginosa* in 115 (48.5%) of the episodes. Carbapenem-resistant bacteria were isolated in 118 (49.8%) samples. Median Pitt Score was 1 (0–4) and 110 (46.4%) patients were admitted to the ICU at baseline. Respiratory tract infections were the most common (40.9%), followed by catheter-related infections (21.1%). Polymyxin B was the most often used antimicrobial, accounting for 115 (48.5%) of the cases. Combination therapy was prescribed for 86 (36.2%) of the infections. The most common antimicrobial combination was of polymyxin B and meropenem, which was prescribed for 73 (84.9%) of these 86 patients.

In total, 67 patients (28.3%) received short-term treatment with a median of six (5–7) days, compared to 170 (71.7%) who received long-term treatment, with a median of 12 (10–15) days of therapy. Table 1 describes general characteristics of the cohort and the univariate analysis comparing baseline variables of short- versus long-term therapy groups. Patients in short-therapy group had significantly more hepatic disease comorbidity (*p* = 0.02), more catheter-related infections (*p* = 0.05), took a longer time to start active therapy (*p* = 0.05), had fewer pulmonary site infections (*p* = 0.02), were less frequently ICU admitted (*p* < 0.01) or on mechanical ventilation (*p* < 0.01) at baseline, were less frequently in septic shock at BSI diagnosis (*p* < 0.01), received less polymyxin B prescription (*p* < 0.01) and had less use of combination therapy (*p* = 0.01) when compared to patients in the long-term therapy group.

### 2.1. Primary Outcome

Thirty-day mortality occurred for 50 (21.1%) of the patients: 8 (11.9%) versus 42 (24.7%) patients treated with short- and long-term therapy, respectively, *p* = 0.03. In multivariate analysis, no statistically significant difference was shown between short- and long-term therapy for 30-day mortality, adjusted hazard ratio (aHR) 1.01, 95% confidence interval (CI) 0.47–2.20, *p* = 0.98. Variables included in the final Cox regression model are shown in Table 2. The propensity score (PS)-adjusted model also did not show impact of short courses of therapy over long courses (aHR 1.16, 95% CI 0.51–0.63, *p* = 0.72). Figure 2 shows adjusted 30-day survival curve of patients receiving short- and long-term therapy courses, with no statistically significant difference between groups, *p* = 0.98.

### 2.2. Secondary Outcomes

During hospitalization, 78 patients (38.9%) died: 11 (16.4%) versus 67 (39.4%) patients in short- vs. long-term therapy groups, respectively, *p* < 0.01. In multivariate analysis, short-term therapy was not a risk factor for in-hospital mortality (Table 2).

Patients survived hospitalization and received hospital discharge in 159 (67.1%) out of the 237 BSI episodes. Median time to hospital discharge after BSI was 12 (7.25–26.75) versus 21 (14–36) days in the short- and long-term therapy groups, respectively, *p* < 0.01.

Within 30 days of the first BSI episode, blood cultures from 133 (56.1%) patients were collected. Among these, 25 (18.8%) of the follow-up blood cultures were positive for the same bacteria: four (13.8%) of 29 in the short-term therapy group versus 21 (20.2%) of 104 in the long-term therapy group, *p* = 0.59. Median time to bacteria isolation was 14.5 (4–26.5) days in the short-term therapy group and 10 (2.5–21.5) days in the long-term therapy group, *p* = 0.45.

### 2.3. Subgroup Analyses

One hundred and eighteen (49.8%) BSI episodes were caused by carbapenem-resistant bacteria, from which, 34 (28.8%) were *Pseudomonas aeruginosa* and 84 (71.2%) were ABC. Thirty-day mortality occurred in 40 (33.9%) of the 118 patients with carbapenem-resistant infections versus 10 (8.4%) of the 119 patients with carbapenem-susceptible ones, *p* < 0.01. In the subgroup of patients with infections caused by carbapenem-resistant bacteria, 30-day mortality occurred in 6 (22.2%) of 27 versus 34 (37.4%) of 91 patients in short- and long-term therapy groups, respectively, *p* = 0.17. Short-term therapy was not an independent risk factor for 30-day mortality, aHR 0.87, 95% CI 0.36–2.11, *p* = 0.76, when adjusted for Charlson Comorbidity Index score (*p* = 0.02) and Pitt bacteraemia score (*p* = 0.06).

In the subgroup of 122 (51.5%) patients with ABC infections, 33 (27.0%) patients died within 30 days of BSI: 6 (18.2%) of 33 versus 27 (30.3%) of 89 in short- and long-term therapy groups, respectively, *p* = 0.25. Short-term therapy was not independently associated with 30-day mortality (aHR1.27, 95% CI 0.51–3.20, *p* = 0.61) when adjusted for Pitt bacteraemia score, Charlson Comorbidity Index score, and carbapenem resistance.

From 115 (48.5%) patients with *Pseudomonas aeruginosa* infections, 17 (14.8%) patients died within 30-days of BSI: two (5.9%) of 34 vs. 15 (18.5%) of 81 in short- and long-term therapy groups, respectively, *p* = 0.09. Short-term therapy was not independently associated with 30-day mortality (aHR 0.52, 95% CI 0.12–2.31, *p* = 0.39), when adjusted for Pitt bacteraemia score, Charlson Comorbidity Index score, and carbapenem resistance.

## 3. Discussion

In our study, antimicrobial therapy duration longer than 7 days was not associated with a lower 30-day mortality rate in patients with *Pseudomonas aeruginosa* or ABC BSI compared to shorter therapy courses in a multivariate COX regression model or in a PS-adjusted model. Therapy duration also did not affect in-hospital mortality. For patients who received hospital discharge, the short-course therapy was associated with a significantly shorter hospital stay. Short-term therapy had microbiological eradication rates comparable to longer therapy periods, however only 56.1% of the patients had control cultures and these cultures were collected in different time periods, which might affect interpretation of these data.

Previous studies have evaluated ideal treatment duration for Gram-negative BSI. The largest study performed to date was a randomized clinical trial with 604 patients showing uncomplicated Gram-negative BSI, which found that a 7-day course of therapy was similar to a 14-day course, for 90-day mortality. This study found no difference in infection recurrence and hospital readmission [13]. However, *E. coli* caused 60.8% of the BSI, mostly secondary to urinary site infections. *Pseudomonas aeruginosa* corresponded to 9.2% and *Acinetobacter baumannii* 0.7% of the cases. Unlike our study, most patients had non-serious infections with a low incidence of carbapenem-resistant bacteria [13]. Another recent randomized clinical trial comparing 7- vs. 14-day therapy for BSI showed non-inferiority of short-term regimens; however, only *Enterobacterales* infections were evaluated in this study [14].

Patients with BSI caused by carbapenem-resistant organisms accounted for half of our sample and had higher mortality rates than the ones with carbapenem-susceptible infections. Carbapenem-resistance was an independent risk factor for 30-day and in-hospital mortality, which is in accordance with the current literature [15,16]. In the subgroup analysis of these patients, results went in the same direction of the main model, not showing benefit of longer therapy courses.

Non-fermenting Gram-negative bacilli are less frequent causes of BSI when compared to *Enterobacterales* species; nevertheless, they deserve special attention due to their peculiar mechanisms of virulence and pathogenicity, intrinsic resistance to multiple drugs, higher rates of acquired multidrug resistance, and association with more severe infections [17,18]. Although those are common aspects between non-fermenting Gram-negative bacilli, it is important to highlight some important differences between *Pseudomonas aeruginosa* and ABC bacteria. For example, the use of potent toxins or virulence factors to perpetuate bacteraemia is more common in *Pseudomonas aeruginosa*, while ABC survival relies mostly on environmental adaptation ability and capsule biosynthesis to protect from opsonization. Moreover, a recent prospective cohort found that the impact of carbapenem resistance in the mortality outcome was worse in patients with ABC BSI when compared to *Pseudomonas aeruginosa* BSI [19]. Therefore, we also evaluated *Pseudomonas aeruginosa* and ABC infections separately, but longer therapy regimens did not show survival protection in either of the groups.

Only a few studies have included non-fermenting, carbapenem-resistant, and critically ill patients. Three retrospective cohort studies evaluating the effect of treatment duration on 30-day mortality and infection recurrence for *Pseudomonas aeruginosa* BSI found no statistically significant difference between patients treated with short-term therapy (median of 8, 8, and 9 days) versus long-term therapy (median of 13, 15, and 16 days) [9,10,11]. Although the definitions of short-term treatment were different from our study, our results followed similar paths, with no difference in 30-day outcomes regarding therapy duration. Recently, a study evaluating treatment duration (7 days versus 14 days) for carbapenem-resistant *Acinetobacter baumannii* BSI in cancer patients found that longer therapies were associated with lower 30-day mortality and better microbiological response rates. All patients had received colistin treatment [12]. However, that study focused on immunosuppressed patients with a cancer diagnosis, which may explain the difference in results from what was found in our cohort.

Our study has limitations due to its retrospective design. First, information regarding probable primary infection site was retrieved from medical records according to the medical assistant evaluation, which might have some heterogeneity. Nevertheless, all patients had confirmed BSI, which assures that patients had clinically relevant infections. Second, this type of study has two main possible sources of bias, which are indication and survival bias [20]. In the first case, longer courses of therapy may be chosen for patients with more severe diseases, whereas ending therapy within 7 days is most probable in patients with a favourable clinical response [20]. We tried to minimize the risk of indication bias by evaluating distinct factors related to disease severity and controlling it in the multivariate model. In fact, in univariate analysis, patients in long-term therapy groups had more severe disease at baseline; however, no difference in mortality was found in the multivariate model when adjusting for Pitt bacteraemia score, Charlson Comorbidity Index score, and carbapenem resistance. In the second case, survival bias, it is possible that some patients received longer courses of therapy because they lived longer and not the other way around [20]. To overcome survival bias, we only analysed patients who were alive on the 8th day after the beginning of active therapy. At that point, patients were allocated in either short- or long-term therapy groups according to the assistant physicians’ decision about stopping or continuing therapy. Mortality was measured from then on, with group allocation already defined. Although excluding patients who died before the 8th day of active therapy was necessary to avoid survival bias, it reduced our sample and compromised, in part, the study power.

In addition to the retrospective design, we had broad heterogeneity of antimicrobial treatments and doses prescribed. New antibiotics such as cefiderocol or combinations of beta-lactam/beta-lactamase inhibitors (e.g., ceftazidime-avibactam, meropenem-varbobactam) were either not available in Brazil during the study period or had their use limited due to high costs. We could expect better outcomes if patients had been treated with these new antimicrobials compared to old antimicrobials, such as polymyxins and aminoglycosides, especially for carbapenem-resistant infections. Nevertheless, no benefit was shown with longer courses of therapy, even when using these old antimicrobials. Finally, we chose the cut-off point of seven days to define short- and long-term therapy because most studies have used it before and would be a clinically relevant division. However, there was a variation in treatment time within each group and this may have had an impact on patient’s outcome. We tried to account for other potential imbalances among groups in multivariate analysis.

## 4. Materials and Methods

### 4.1. Study Designs and Settings

This was a retrospective cohort study carried out from October 2013 to October 2019 in two tertiary-care teaching hospitals in Porto Alegre, Brazil: hospital A with 580 beds and hospital B with 919 beds.

### 4.2. Inclusion and Exclusion Criteria

Patients aged ≥ 18 years with BSI caused by ABC or *Pseudomonas aeruginosa* treated with in vitro-susceptible antimicrobials were included. Patients were excluded if: ≤48 h treatment duration, previous BSI episode by the same bacteria in ≤30 days, complicated BSI (defined by the persistence of infection foci, such as endocarditis, osteomyelitis, and cavity abscesses), polymicrobial infections (growth of more than one microorganism in blood cultures), and febrile neutropenia. Also, patients who died within <8 days of the beginning of active antimicrobial therapy were excluded, for the choice between short-term therapy or prolonged antimicrobial regimens is only possible in patients who survived this initial period. From that point on, mortality was compared between groups, assuring that group allocation was not driven by patients’ survival status.

### 4.3. Variables and Definitions

The primary outcome was 30-day mortality in patients with *Pseudomonas aeruginosa* or ABC BSI. Secondary outcomes were in-hospital mortality, microbiological eradication within 30 days, and time until hospital discharge. The main independent variable was therapy duration, categorized based on short-term (≤7 days) and long-term (>7 days) therapies.

We performed a query to screen for BSI caused by *Pseudomonas aeruginosa* and ABC recovered in our microbiology laboratory during the study period. Patients were included in the first day of bacteraemia, defined as the day when the blood sample was collected. Our institutional protocol recommends the collection of at least two blood culture samples per patient at the discretion of the medical assistant. Bacterial identification and antimicrobial susceptibility tests were performed using the Vitek2^®^ automatized system or MALDI-TOF (bioMérieux, France) system. Polymyxin susceptibility was evaluated by broth microdilution tests. Results were interpreted according to the Clinical and Laboratory Standards Institute (CLSI 2014) guidelines [21].

Baseline variables potentially related to outcomes were assessed by the researchers through review of the medical records: demographic variables (age, gender), comorbidities (underlying diseases of patients and the Charlson Comorbidity Index score [22]), primary infection site according to the patient’s medical record (respiratory, abdominal, urinary, skin and soft tissue, catheter-related BSI [6], and undefined site), ICU admission, invasive mechanical ventilation, haemodialysis, Pitt bacteraemia score [23], septic shock, isolated bacteria (*Pseudomonas aeruginosa* or *ABC*), carbapenem resistance, time to start active therapy according to medical prescription (active therapy was defined as an antimicrobial for which the isolated bacteria had in vitro susceptibility), combination therapy (defined as the association of two or more antimicrobials), and removal or retention of central venous catheter (in catheter-related infections).

The follow-up period of patients was defined as the date from BSI to hospital discharge. Patients who received hospital discharge before 30 days of follow-up were not contacted thereafter and were censored in the analysis at that point.

### 4.4. Sample Size

We planned a study of independent groups, expecting a 2:1 ratio of patients receiving long- and short-term therapy. In the study by Katip et al., which evaluated carbapenem-resistant ABC BSI, the overall mortality was 28.9%, and long-course therapy was protective, with a HR of 0.11, compared to that for short-course therapy [12]. To estimate our sample, we considered a more conservative HR of 0.5 (5 times higher than the previous study, but still clinically significant) and assumed a mortality rate of 30% and 40% among long- and short-term therapy groups, respectively. A minimum of 189 subjects (169 in the long-term therapy group and 63 in the short-therapy group) would have to be recruited to be able to reject the null hypothesis with a power of 0.8. and a type I error probability of 0.05. Adding 20% in this sample for possible losses and multivariate analysis, we calculated a minimum of 227 patients to be included [24].

We did a *post- hoc* analysis with the actual number of recruited patients in each group and considering the 30-day mortality rate found in our study, which led to a study power of 70.4% [24].

### 4.5. Statistical Analysis

Statistical analyses were performed using SPSS for Windows, Version 18.0. We calculated the median and percentiles (*p*) 25th and 75th (p25–p75) for ordinal or non-normally distributed variables, mean and standard deviation (SD) for normally distributed variables, and total and percentage value for categorical variables. Bivariate analysis was performed separately for each of the variables to evaluate differences between short- and long-term therapy groups. The *p* values were calculated using Fisher’s exact test for categorical variables, and Student’s *t*-test or the Wilcoxon–Mann–Whitney test for continuous variables. All tests were two-tailed, and a *p* value ≤ 0.05 was considered significant.

A Cox regression model was used to assess the effect of antibiotic therapy duration on 30-day mortality, adjusting for other potential confounding variables. Survival analyses were used to censor patients at hospital discharge (when it occurred before 30 days) or in the end of follow-up. Variables with *p* < 0.20 in the bivariate analysis were included, one by one, in a stepwise-forward model, starting with those with lower *p* values. If equal *p* values occurred, the variable with the greatest magnitude of effect was included first. Variables with *p* < 0.05 were retained in the model. We also performed a propensity score (PS)-adjusted analysis. Variables included in the PS were age, sex, isolated bacteria, infection site, Charlson comorbidities, Pitt bacteraemia score, hospital of admission, ICU admission, mechanical ventilation, and septic shock at baseline.

Secondary outcomes were in-hospital mortality, microbiological eradication within 30 days, and time to hospital discharge, which were also evaluated in a COX regression model, adjusting for the same variables of the main multivariate model. A pre-planned subgroup analysis was conducted for carbapenem-resistant infections and separately for ABC and *Pseudomonas aeruginosa*.

## 5. Conclusions

Our study demonstrated that long-term therapies (>7 days) for *Pseudomonas aeruginosa* and ABC BSI were not superior to short-term therapies (≤7 days) for 30-day mortality, when controlled for Charlson Comorbidity Index score, Pitt bacteraemia score, and carbapenem resistance. Short-term therapies were associated with shorter hospital stay. Although randomized clinical trials evaluating therapy duration for *Pseudomonas aeruginosa* and ABC BSI are not available, our data suggest that short-course therapies are safe for uncomplicated BSI caused by these organisms.

## Figures and Tables

**Figure 1 antibiotics-12-00538-f001:**
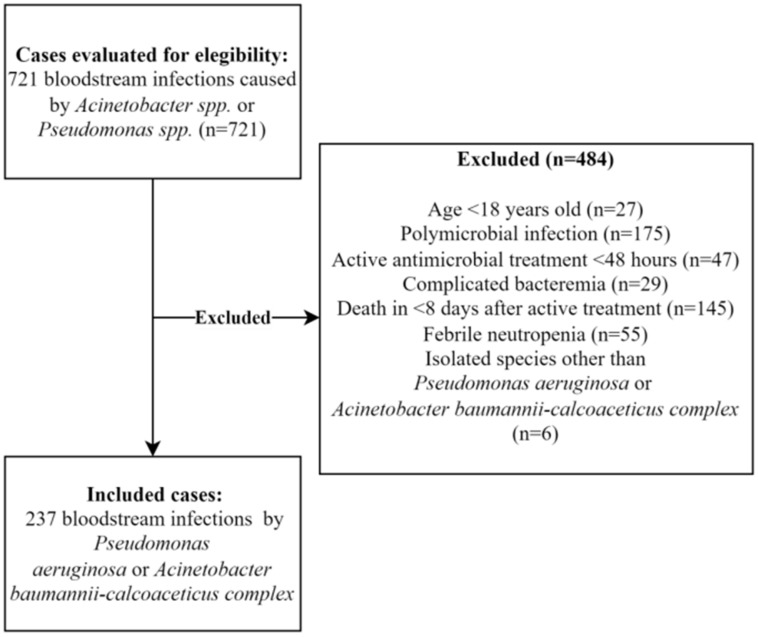
Study inclusion flowchart.

**Figure 2 antibiotics-12-00538-f002:**
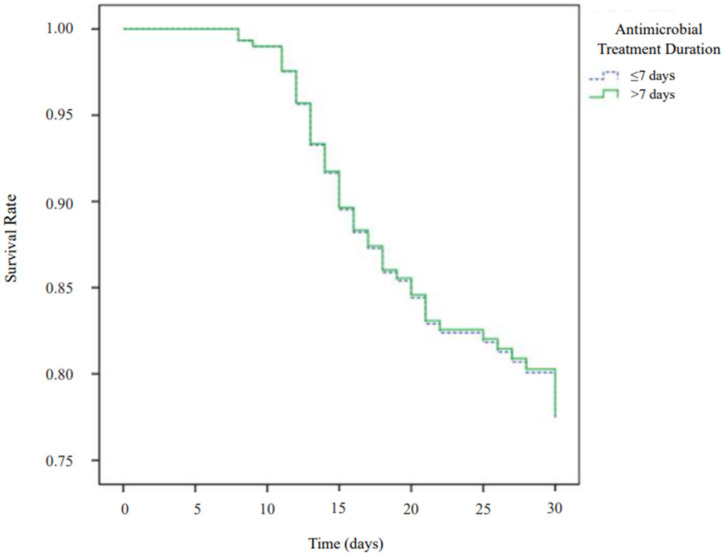
Cox survival analyses comparing 30-day mortality between short- versus long-term therapy for *Pseudomonas aeruginosa* or *Acinetobacter baumannii-calcoaceticus complex* bloodstream infections.

**Table 1 antibiotics-12-00538-t001:** Baseline characteristics and univariate analysis comparing short- and long-term therapy groups in patients with *Acinetobacter baumannii-calcoaceticus complex* and *Pseudomonas aeruginosa* bloodstream infection.

Characteristic	Total Cohort	Therapy Duration		
	N = 237	≤7 Days, N = 67	>7 Days, N = 170	*p* Value
** *Demographic* **				
Age	61.6 ± 16.0	60.5 ± 15.8	62.0 ± 16.1	0.50
Gender (masculine)	122 (51.5)	34 (50.7)	88 (51.8)	0.99
** *Comorbidity* **				
Cardiovascular	152 (64.1)	46 (68.7)	106 (62.4)	0.45
Pulmonary	66 (27.8)	21 (31.3)	45 (26.5)	0.52
Neurological	90 (38.0)	31 (46.3)	59 (34.7)	0.11
Hepatic	13 (5.5)	0 (0)	13 (7.6)	0.02
Chronic kidney disease	70 (29.5)	19 (28.4)	51 (30.0)	0.88
Gastrointestinal	41 (17.3)	11 (16.4)	30 (17.6)	0.99
Diabetes	64 (27.0)	14 (20.9)	50 (29.4)	0.20
HIV	16 (6.8)	5 (7.5)	11 (6.5)	0.78
Rheumatic	8 (3.4)	3 (4.5)	5 (2.9)	0.69
Solid organ cancer	55 (23.2)	11 (16.4)	44 (25.9)	0.13
Haematological cancer	8 (3.4)	4 (6.0)	4 (2.4)	0.23
Charlson Comorbidity Index	5 (3–7)	5 (2–7)	5 (3–8)	0.09
** *Hospital of Admission* **				
Hospital A	215(90.7)	57(85.1)	158(92.9)	0.08
** *Disease severity* **				
ICU	110 (46.4)	22 (32.8)	88(51.8)	<0.01
Mechanical ventilation	64 (27.0)	10 (14.9)	54 (31.8)	<0.01
Pitt bacteraemia score	1 (0–4)	1 (0–2)	2 (0–6)	0.10
Septic shock	46 (19.4)	5 (7.5)	41 (24.1)	<0.01
** *Microbiological data* **				
*ABC* infections	122 (51.5)	33 (49.3)	89 (52.4)	0.77
*Pseudomonas aeruginosa* infections	115 (48.5)	34 (50.7)	81 (47.6)	0.77
Multidrug resistance	115 (48.5)	26 (38.8)	89 (52.4)	0.06
Carbapenem resistance	118 (49.8)	27 (40.3)	91 (53.5)	0.08
Time from hospitalization to bacteraemia (days)	13 (3.0–29.8)	16 (4–27)	12 (2–30.5)	0.76
** *Infection site* **				
Pulmonary	97 (40.9)	19 (28.4)	78 (45.9)	0.02
Urinary	36 (15.2)	12 (17.9)	24 (14.1)	0.55
Abdominal	19 (8.0)	5 (7.5)	14 (8.2)	0.99
Central venous catheter	50 (21.1)	20 (29.9)	30 (17.6)	0.05
Removal of CVC (up to 48 h)	34 (68.0)	12 (60.0)	22 (73.3)	0.35
Skin and soft tissues	19 (8.0)	8 (11.9)	11 (6.5)	0.19
Undefined site	33 (13.9)	10 (14.9)	23 (13.5)	0.84
** *Antimicrobial treatment* **				
Ampicillin-sulbactam *	21 (8.9)	7 (10.4)	14 (8.2)	0.62
Ceftazidime	26 (11.0)	7 (10.4)	19 (11.2)	0.99
Cefepime	15 (6.3)	5 (7.5)	10 (5.9)	0.77
Piperacillin-tazobactam	52 (21.9)	17 (28.4)	33 (19.4)	0.16
Ciprofloxacin	35 (14.8)	13 (19.4)	22 (12.9)	0.23
Amikacin	3 (1.3)	1(1.5)	2 (1.2)	0.99
Meropenem	52 (21.9)	10 (14.9)	42 (24.7)	0.19
Polymyxin B	115 (48.5)	22 (32.8)	93 (54.7)	<0.01
Colistin	4 (1.7)	2 (3.0)	2 (1.2)	0.32
Time to start active antimicrobial therapy (days)	1 (0–2)	1 (0–2)	0 (0–2)	0.05
Combination therapy	86 (36.2)	16 (23.9)	70 (41.2)	0.01
Polymyxin B + meropenem	73 (30.8)	13 (19.4)	60 (35.3)	
Polymyxin B + amikacin	4 (1.7)	1 (1.5)	3 (1.8)	
Polymyxin B + ampicillin-sulbactam	2 (0.84)	0	2 (1.2)	
Polymyxin B + other antibiotics	3 (1.3)	0	3 (1.8)	
Colistin+ other antibiotics	4 (1.7)	2 (3.0)	2 (1.2)	

ABC, *Acinetobacter baumannii-calcoaceticus complex;* CVC, central venous catheter; ICU, intensive care unit admission. * Ampicillin-sulbactam was used only for ABC infections when activity of this drug was shown in the antibiogram. Results are presented as: mean± standard deviation, median (interquartile range) or *n* (%)

**Table 2 antibiotics-12-00538-t002:** Multivariate analysis for 30-day and in-hospital mortality in patients with Pseudomonas aeruginosa or Acinetobacter baumannii-calcoaceticus complex bloodstream infections.

Variables	30-Day Mortality			In-Hospital Mortality		
	aHR	95% CI	*p*	aHR	95% CI	*p* Value
Short-term therapy (vs. long-term) **	1.01	0.47–2.20	0.98	0.70	0.37–1.34	0.70
Pitt bacteraemia score	1.10	1.02–1.19	0.02	1.08	1.01–1.15	0.03
Charlson Comorbidity Index	1.18	1.07–1.30	<0.01	1.13	1.04–1.22	<0.01
Carbapenem resistance	2.65	1.28–5.48	<0.01	1.91	1.09–3.36	0.02

aHR, adjusted hazard ratio; CI, confidence interval. ** Short term therapy ≤ 7 days, long-term therapy > 7 days.

## Data Availability

Unidentified datasets will be made available by the corresponding author on reasonable request.

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
