# Peer review of "Antimicrobial Therapy Duration for Bloodstream Infections Caused by Pseudomonas aeruginosa or Acinetobacter baumannii-calcoaceticus complex: A Retrospective Cohort Study"

_antibiotics, 2023, doi:10.3390/antibiotics12030538_

Round 1
Reviewer 1 Report
1、"P" should be italicized.
2、Numbers should be written in a uniform format. Two formats appear in the text, such as "During hospitalization, 78 patients (38.9%) died" ,"One-hundred and eighteen (49.8%) BSI episodes".
3、As a retrospective study, the sample size of this study is too small. Is the sample size enough to reach this conclusion? Please explain. Alternatively, please explain the sample size estimation method for this retrospective study.
Author Response
Dear Editor,
Thank you for considering our manuscript for publication in Antibiotics and for the important comments from reviewers, which were addressed in the new version of the manuscript.
Please find the author’s response (AR) to each of the comments below.
Yours sincerely,
Maria Helena Rigatto
Reviewer 1
1、"P" should be italicized.
AR: We thank the reviewer for this remark, we changed P values to italics in the manuscript.
2、Numbers should be written in a uniform format. Two formats appear in the text, such as "During hospitalization, 78 patients (38.9%) died","One-hundred and eighteen (49.8%) BSI episodes".
AR: We thank the reviewer for pointing this out. We adjusted it in the new version of the manuscript.
3、As a retrospective study, the sample size of this study is too small. Is the sample size enough to reach this conclusion? Please explain. Alternatively, please explain the sample size estimation method for this retrospective study.
AR: We thank the reviewer for this important consideration. We had planned a study of independent groups, expecting a 2:1 ratio of patients receiving long- and short-term therapy. In the study of Katip et al, which evaluated carbapenem-resistant ABC BSI, the overall mortality was of 28.9% and long-course therapy was protective with a HR of 0.11 compared to short-course therapy [9]. To estimate our sample, we considered a HR of 0.5 (5 times higher, but still clinically significant) and assumed a mortality rate of 30% and 40% among long- and short-term therapy groups, respectively. A minimum of 189 subjects (169 in the long therapy group and 63 in the short-therapy group) would have to be recruited to be able to reject the null hypothesis with power of 0.8. and type I error probability of 0.05. Adding 20% in this sample for possible losses and multivariate analysis, we calculated a minimum of 227 patients to be included.
We did a post- hoc analysis with the actual number of recruited patients in each group and considering the 30-day mortality rate found in our study, which led to a study power of 70.4%.
We added this information in lines 301-312 of the new version.
Author Response
Dear Editor,
Thank you for considering our manuscript for publication in Antibiotics and for the important comments from reviewers, which were addressed in the new version of the manuscript.
Please find the author’s response (AR) to each of the comments below.
Yours sincerely,
Maria Helena Rigatto
Reviewer 2
- The authors did not clarify the exact duration of antibiotic therapy, simplifying it in <7 or >7 days. I think that it may be useful to report the exact duration. Also, some dosages would be useful.
AR: We thank the reviewer for this comment. Median therapy duration was of six (5–7) days and 12 (10–15) days in short- and long-term therapy groups, respectively. This information is described in lines 93-94 of the new version. Because there were many antimicrobials used, sometimes with dose adjustments for renal function, we did not specifically analyze dose for this work. We added this on limitations lines 252-253 of the new version.
- I suggest to better explain the antibiotic management of patients described.
Which combination was the most used? Is there significative difference in terms of combination?
AR: We thank the reviewer for this suggestion. The most frequent combination was of polymyxin B and meropenem for 73(30.3%) of the patients. We added that information in lines 86-88 and described the combinations in table 1.
- Why did you not mention newer antibiotics, such as cefiderocol, ceftazidime-avibactam, meropenem-vaborbactam, ceftolozane-tazobactam, imipenem-relebactam?
Please explain this point. Moreover, please read and evaluate those two papers
https://doi.org/10.3390/antibiotics12010049 and 10.1128/spectrum.02347-22
AR: We thank the reviewer for this important comment and for sharing the articles. Unfortunately, most of these new antibiotics are not available in Brazil, including cefiderocol and intravenous fosfomycin. From the new antibiotics launched, ceftazidime-avibactam is the only available in Brazil. It was approved by our national regulation agency in 2018, but still has its use restricted due to high costs. We added this aspect in our limitations, lines 252-257.
- Please add more reference and cite more study about therapy duration in those patients.
AR: We thank the reviewer for this suggestion. We added the following references in the new version of our work:
- Ruiz-Ruigómez M, Aguado JM. Duration of antibiotic therapy in central venous catheter-related bloodstream infection due to Gram-negative bacilli. Curr Opin Infect Dis. 2021 Dec 1;34(6):681-685. doi: 10.1097/QCO.0000000000000763.
- Molina J et al. Seven-versus 14-day course of antibiotics for the treatment of bloodstream infections by Enterobacterales: a randomized, controlled trial. Clinical Microbiology and Infection (2022) 28(4) 550-557 DOI: 10.1016/j.cmi.2021.09.001
- Babich T. Duration of Treatment for Pseudomonas aeruginosa Bacteremia: a Retrospective Study. Infect Dis Ther. 2022 Aug;11(4):1505-1519. doi: 10.1007/s40121-022-00657-1.
- Almeida Junior ER, et al. Multicentre surveillance of epidemiologically important pathogens causing nosocomial bloodstream infections and pneumonia trials in Brazilian adult intensive care units. J Med Microbiol. 2023 Feb;72(2). doi: 10.1099/jmm.0.001654.
- Line 60: “This subject” please change
AR: We thank the reviewer for pointing this out. We changed the sentence to “ Therapy duration becomes more controversial in…” (line 64 of the new version).
- Table 2: Please adjust since it is unclear
AR: We thank the reviewer for this comment, we adjusted table 2 to a clearer format.
- Line 151: “30-day” what?
AR: We thank the reviewer for this remark. We rephrased the sentence to “…patients died within 30 days of BSI” (lines 176-177 of the new version)

Reviewer 3 Report
The current study, "Antimicrobial therapy duration for bloodstream infections caused by Pseudomonas aeruginosa or Aci-2 Acinetobacter baumannii-calcoaceticus complex," provides information on retrospective cohort study evaluating the effect of treatment duration on 30-day mortality and infection recurrence for Pseudomonas aeruginosa BSI, where the result is compared for short (≤7 days) and long-term (>7 days) antimicrobial therapy duration for Pseudomonas aeruginosa infections.
Comments:
1. The introduction is short, rough and difficult to understand, it needs to be focused more on Pseudomonas aeruginosa. If possible please add in which age group this infection generally seen, where it is found, what is the prevalence rate in Brazil, etc.
2. The introduction seems to be unclear and unspecific. Please cross-check the percentage of BSI cases provide in the manuscript. Is the citation provided in the manuscript correct?
3. Proper citation is not provided and please cross-check the references.
4. The text needs extensive proofreading in English and there are many typing errors that should be taken care of. For eg. Spp., ABC cannot be in italics.
5. The result section seems to be chaotic and difficult to understand, for the following reasons:
As result, it is mentioned that 122 patients were male. Is the study done only in males? In table 1, what does P stands for? 2.1 Primary Outcome- difficult to comprehend? Revise this section to include more information about Table 2 and Figure 2. Specify which bacteria are responsible for the 118 carbapenem-resistant bacteria.
Add some percentage comparisons from your study to provide a clear overview in the discussion part of the manuscript. The technique was performed to find carbapenem-resistant bacteria. The conclusion seems very sketchy and short- Make it clearer and more comprehensive. Authors should try to cite the paper with proper referencing.
Author Response
Dear Editor,
Thank you for considering our manuscript for publication in Antibiotics and for the important comments from reviewers, which were addressed in the new version of the manuscript.
Please find the author’s response (AR) to each of the comments below.
Yours sincerely,
Maria Helena Rigatto
Reviewer 3
The current study, "Antimicrobial therapy duration for bloodstream infections caused by Pseudomonas aeruginosa or Aci-2 Acinetobacter baumannii-calcoaceticus complex," provides information on retrospective cohort study evaluating the effect of treatment duration on 30-day mortality and infection recurrence for Pseudomonas aeruginosa BSI, where the result is compared for short (≤7 days) and long-term (>7 days) antimicrobial therapy duration for Pseudomonas aeruginosa infections.
Comments:
- The introductionis short, rough and difficult to understand, it needs to be focused more on Pseudomonas aeruginosa. If possible please add in which age group this infection generally seen, where it is found, what is the prevalence rate in Brazil, etc.
AR: We thank the reviewer for this comment. We added more data about Pseudomonas aeruginosa and Acinetobacter baumannii-calcoaceticus complex bloodstream infections in the introduction (lines 56-59).
- The introductionseems to be unclear and unspecific. Please cross-check the percentage of BSI cases provide in the manuscript. Is the citation provided in the manuscript correct?
AR: We thank the reviewer for this remark. We cross-checked the reference provided, and the data is correct. Those percentages are presented on table 2 of the cited article (Diekema et al, 2019).
- Proper citation is not provided and please cross-check the references.
AR: We thank the reviewer for the comment. We adjusted the references to the journal format.
- The text needs extensive proofreading in English and there are many typing errors that should be taken care of. For eg. Spp., ABC cannot be in italics.
AR: We thank the reviewer for this comment. We reviewed the article and corrected these typing mistakes.
- The resultsection seems to be chaotic and difficult to understand, for the following reasons:
5.1 As result, it is mentioned that 122 patients were male. Is the study done only in males.
AR: We thank the reviewer for these remarks. We rephrased the sentence in the results section to “Patients had a mean age of 61.6 ± 16.0 years and 122 (51.5%) of 237 were male” in lines 80-81
5.2 In table 1, what does P stands for?
AR: Table 1 shows the univariate analysis comparing baseline variables between patients in the short- and long-term therapy groups. To be clearer we rephrased the title of table 1 to “Baseline characteristics and univariate analysis comparing short and long-term therapy groups in patients with Acinetobacter baumannii-calcoaceticus complex and Pseudomonas aeruginosa bloodstream infection.” and changed “P” for “P value”.
5.3 Primary Outcome- difficult to comprehend? Revise this section to include more information about Table 2 and Figure 2.
AR: We thank the reviewer for this comment, we adjusted the “primary outcome section” adding more information about table 2 and figure 2 in lines 124-130 of the new version.
5.4 Specify which bacteria are responsible for the 118 carbapenem-resistant bacteria.
AR: We thank the reviewer for this suggestion. We added the sentence “Carbapenem-resistant bacteria accounted for 118 (49.8%) of the BSI episodes, from which 34 (28.8%) were Pseudomonas aeruginosa and 84 (71.2%) were ABC.” In lines 168-169.
- Add some percentage comparisons from your study to provide a clear overview in the discussion part of the manuscript.
AR: We thank the reviewer for the suggestion. We have added some percentages comparison in the results section and discussion (lines 169-171 and 191), to clarify the text.
- The technique was performed to find carbapenem-resistant bacteria.
AR: Bacterial identification and antimicrobial susceptibility tests were performed using Vitek2® automatized system or MALDI-TOF (bioMérieux, France) system. Results were interpreted according to the Clinical and Laboratory Standards Institute (CLSI 2014). This information is cited in lines 284-287 of the new version.
- The conclusion seems very sketchy and short- Make it clearer and more comprehensive.
AR: We thank the reviewer for the comment. We added a few more lines (335-340) in the conclusion section of the manuscript to make it more comprehensive.
- Authors should try to cite the paper with proper referencing
AR: We thank the reviewer for this comment. We checked and adjusted the references.

Reviewer 4 Report
General comments
=============
Thank you for letting me peer-review your work! This paper is an interesting study for Antimicrobial therapy duration for bloodstream infections caused by Pseudomonas aeruginosa or Acinetobacter baumannii-calcoaceticus complex. The manuscript was well written. However, the authors should clarify the exclusion cases and the missing value related to the primary outcome.
Specific comments
=============
Major comments
---------------------
1. More than half cases of pseudomonas or ABC BSI were excluded(484/721). Please discuss these too many excluded cases affecting the result. For example, the exclusion of the patient who died in < 8 days after active treatment would be related to the primary outcomes.
2. Total 237 patients were included. However, 30 days mortality was evaluated only during hospitalization. Additionally, the duration of hospitalization was significantly different between the short- and long-term therapy group. The authors should clarify the missing value for the primary outcome, especially the number of the missing value and how to deal with the missing value.
3. Please clarify the type of this study (retrospective cohort study) in the title.
4. Please add the duration of inclusion in the abstract.
5. Please clarify who checked the medical record, especially the infectious site and time to start active antimicrobial therapy, which was a possibility to interpret discordance among researchers.
5. In general, ampicillin-sulbactam is not effective against the pseudomonas without some special combination therapy. Please discuss it. If there are any special indications, please add them.
6. Please clarify how to perform the blood culture, and how to culture the specimen.
7. If available, please add the power analysis for the primary outcome.
Minor comments
---------------------
8. Please clarify “Hospital A” in table1.
Author Response
Dear Editor,
Thank you for considering our manuscript for publication in Antibiotics and for the important comments from reviewers, which were addressed in the new version of the manuscript.
Please find the author’s response (AR) to each of the comments below.
Yours sincerely,
Maria Helena Rigatto
Reviewer 4
General comments
=============
Thank you for letting me peer-review your work! This paper is an interesting study for Antimicrobial therapy duration for bloodstream infections caused by Pseudomonas aeruginosa or Acinetobacter baumannii-calcoaceticus complex. The manuscript was well written. However, the authors should clarify the exclusion cases and the missing value related to the primary outcome.
Specific comments
=============
Major comments
---------------------
- More than half cases of pseudomonas or ABC BSI were excluded (484/721). Please discuss these too many excluded cases affecting the result. For example, the exclusion of the patient who died in < 8 days after active treatment would be related to the primary outcomes.
AR: We thank the reviewer for this important comment. We added a few lines discussing these aspects in the discussion section: “Although excluding patients who died before the 8th day of active therapy was necessary to avoid survival bias, it reduced our sample and compromised in part the study power.”, lines 249-251
- Total 237 patients were included. However, 30 days mortality was evaluated only during hospitalization. Additionally, the duration of hospitalization was significantly different between the short- and long-term therapy group. The authors should clarify the missing value for the primary outcome, especially the number of the missing value and how to deal with the missing value.
AR: We thank the reviewer for pointing this out. We chose to do a Cox regression model to be able to censor patients of the analysis at hospital discharge, when it occurred before 30-days. We made that explanation clearer in line 324 of the new version of the manuscript.
- Please clarify the type of this study (retrospective cohort study) in the title.
AR: We thank the reviewer for this suggestion. We changed the title to “Antimicrobial therapy duration for bloodstream infections caused by Pseudomonas aeruginosa or Acinetobacter baumannii-calcoaceticus complex: a retrospective cohort study.”
- Please add the duration of inclusion in the abstract.
AR: We thank the reviewer for this suggestion, we have added the inclusion period (2013-2019) in the abstract, line 24.
- Please clarify who checked the medical record, especially the infectious site and time to start active antimicrobial therapy, which was a possibility to interpret discordance among researchers.
AR: We thank the reviewer for this comment. All variables were assessed by the researchers though revision of the medical records and time to start active antimicrobial therapy was retrieved from medical prescription. We made that information clearer in lines 288 and 294, and added it as a limitation in lines 234-237.
- In general, ampicillin-sulbactam is not effective against the pseudomonas without some special combination therapy. Please discuss it. If there are any special indications, please add them.
AR: We thank the reviewer for this remark. Ampicillin-sulbactam was used only for ABC infections when antibiogram showed susceptibility of the bacteria to this drug. This was highlighted in a table 1 footnote of the new version.
- Please clarify how to perform the blood culture, and how to culture the specimen.
AR: We thank the reviewer for this comment, we added this information in lines 283-287.
- If available, please add the power analysis for the primary outcome.
AR: We thank the reviewer for this comment. We added the prior sample size calculation and the post-hoc power analysis in lines 301-312, which was of 70.4%. We mentioned that in the limitations, lines 249-251.
Minor comments
---------------------
- Please clarify “Hospital A” in table1.
AR: We thank the reviewer for this remark. We did not identify the hospital A for ethical reasons, but we have added “Hospital A (versus hospital B)” in table one, to make clearer that we wished to evaluate if there was any significant difference in the proportion of short- versus long-course therapies according to the hospital the patient was included. We also specified that in line 265.

Round 2
Reviewer 4 Report
General comments
=============
I appreciated the opportunity to peer-review your work on Antimicrobial therapy duration for bloodstream infections caused by Pseudomonas aeruginosa or Acinetobacter baumannii-calcoaceticus complex.
The manuscript was written well and presented the research in a clear.Overall, the responses provided by the authors to the reviewer's comments were reasonable and satisfactory.
Author Response
Dear Editor,
Thank you for considering our manuscript for publication in Antibiotics.
Please find the author’s response (AR) to the reviewer comment below.
Yours sincerely,
Maria Helena Rigatto
Reviewer 4
1.
I appreciated the opportunity to peer-review your work on Antimicrobial therapy duration for bloodstream infections caused by Pseudomonas aeruginosa or Acinetobacter baumannii-calcoaceticus complex.
The manuscript was written well and presented the research in a clear.Overall, the responses provided by the authors to the reviewer's comments were reasonable and satisfactory.
AR: We thank the reviewer for this comment. We checked the manuscript for spelling mistakes and corrected it in the new submitted version.